# High Accuracy Pre-Harvest Sugarcane Yield Forecasting Model Utilizing Drone Image Analysis, Data Mining, and Reverse Design Method

**Bhoomin Tanut** [ID]**, Rattapoom Waranusast** [ID] **and Panomkhawn Riyamongkol \*** [ID]

Department of Electrical and Computer Engineering, Faculty of Engineering, Naresuan University, Phitsanulok 65000, Thailand; bhoomint60@nu.ac.th (B.T.); rattapoomw@nu.ac.th (R.W.)
\* Correspondence: panomkhawnr@nu.ac.th; Tel.: +66-087-806-7807

**Abstract:** This article presents a new model for forecasting the sugarcane yield that substantially reduces current rates of assessment errors, providing a more reliable pre-harvest assessment tool for sugarcane production. This model, called the Wondercane model, integrates various environmental data obtained from sugar mill surveys and government agencies with the analysis of aerial images of sugarcane fields obtained with drones. The drone images enable the calculation of the proportion of unusable sugarcane (the defect rate) in the field. Defective cane can result from adverse weather or other cultivation issues. The Wondercane model is developed on the principle of determining the yield not through data in regression form but rather through data in classification form. The Reverse Design method and the Similarity Relationship method are applied for feature extraction of the input factors and the target outputs. The model utilizes data mining to recognize and classify the dataset from the sugarcane field. Results show that the optimal performance of the model is achieved when: (1) the number of Input Factors is five, (2) the number of Target Outputs is 32, and (3) the Random Forest algorithm is used. The model recognized the 2019 training data with an accuracy of 98.21%, and then it correctly forecast the yield of the 2019 test data with an accuracy of 89.58% (10.42% error) when compared to the actual yield. The Wondercane model correctly forecast the harvest yield of a 2020 dataset with an accuracy of 98.69% (1.31% error). The Wondercane model is therefore an accurate and robust tool that can substantially reduce the issue of sugarcane yield estimate errors and provide the sugar industry with improved pre-harvest assessment of sugarcane yield.

**Keywords:** pre-harvest sugarcane yield forecasting model; reverse-design feature extraction; the similarity relationship method; data mining

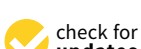



## 1. Introduction

The agricultural sector is a vital part of the Thai economy because 9.37 million people are engaged in agriculture. Currently the five main crops of Thailand, in order of economic importance, are: rice, cassava, sugarcane, rubber, and palm [1]. Sugarcane is one of the main cash crops in Thailand and many researchers are interested in studying sugarcane due to its popularity among the farmers, and its price is also profitable and stable when compared to other local crops like rice, cassava, rubber, and palm. In 2019, Thailand produced a total of 130.91 million metric tons of sugarcane, worth approximatelyUSD 176 million [2] and as of 2019, the country was the world's third largest exporter of sugar after Brazil, and India [3]. In Thailand, approximately 63.63% of the sugarcane fields are harvested not by the farmers, but rather by sugar mills, under an agreement with the farmers [4]. In this case the sugar mill needs to survey the field before harvest in order to estimate the amount of harvesting equipment, human labor, and other resources to allocate for that field. Accurate yield estimation of unharvested sugarcane has long been a great challenge, and this issue has expensive consequences when equipment, labor, and other resources cannot be allocated efficiently. Smart technology for agriculture can provide

solutions. Sensor systems, advanced data analysis techniques, and big data analytics can have a real impact on improving sugarcane yield estimation.

Currently, the forecasting of sugarcane yield relies on expert surveyors whose estimate is based on their own past experiences. The subjectivity of this method makes it prone to assessment errors, which can result, for example, from the expansion of cultivated areas, differences in local environmental conditions, or recent cultivation issues, or natural events such as adverse weather [5]. Thus, sugar mills seek standardized and robust tools that can assess sugarcane yield more accurately and reliably. At present, various agricultural models have been developed to forecast crop yields, and these models can generally be divided into two types: (1) models based solely on environmental data, and (2) models based on environmental data as well as images. Among the first type, Buket et al. [6], presented a sugarcane yield forecasting model that uses an artificial neural network. Hammer et al. [7], developed a sugarcane yield forecasting model that uses the Random Forest algorithm together with Crop Simulation. Srikamdee et al. [8] created a sugarcane yield forecasting model using deep learning. Among the second type of model that has been previously developed (models incorporating both environmental data and images) some use satellite images and others use drone images.

The following studies utilize satellite images. Adisa et al. [9] presented a corn yield forecasting model that uses a vegetation index calculated from satellite images together with an artificial neural network. Prathumchai et al. [10] created a sugarcane yield forecasting model that employs the Leaf Area Index (LAI), also based on satellite images, and Rahman et al. [11] developed a sugarcane yield forecasting model utilizing the satellite image-based green normalization vegetation index (GNDVI).

Previously introduced models that use drone images include that of Matese et.al. [12] whose analyses of plant diseases in vineyards use drone images based on NDVI. Mink et al. [13] developed a weed detection model for maize and sugar beet using drone images based on GNDVI. RGB color images from a drone are combined with environmental data to develop the pre-harvest sugarcane yield forecasting model of Som-ard et al. [14] That model uses Object-Based Image Analysis (OBIA) with a Gray-Level Co-occurrence Matrix (GLCM). Sanches et al. [15] created a sugarcane yield forecasting model that uses a Green–Red Vegetation Index (GRVI) and a Leaf Area Index (LAI) that are both drone image based. Both of the studies by Som-ard et al. [14] and and Sanches et al. [15] demonstrated how RGB color image analysis can be applied to estimate sugarcane yield. Som-ard et al.'s study [14] resulted in a highly accurate (OA) estimate (92% accuracy) for a small number of fields with a large total area (2 fields with a total area of approximately 36,000 square meters). On the other hand, Sanches et al.'s study [15] used more fields with a smaller total area (15 fields with a total area of approximately 1800 square meters), and its estimate accuracy (also OA) was lower (90% accuracy). The methods of Som-ard et al. [14] and Sanches et al. [15] can be applied to forecast the yield of sugarcane fields in some parts of Thailand.

Kamphaeng Phet Province is one of the Thai provinces where sugarcane is most widely cultivated. It is located the Northern Region of Thailand (Latitude: 16.4183581 Longitude: 99.6111616). The total area of the province is 8512 km$^2$, and its sandy soil is suitable for growing a variety of crops. The major agricultural crops of Kamphaeng Phet Province are rice, cassava, maize, sugarcane, banana, and tobacco. The climate in Kamphaeng Phet is characterized by high temperatures throughout the year, with alternating rain and drought, as well as storms that cause damage to agricultural crops. Annual temperatures and annual rainfall of Kamphaeng Phet are shown in Figure 1.

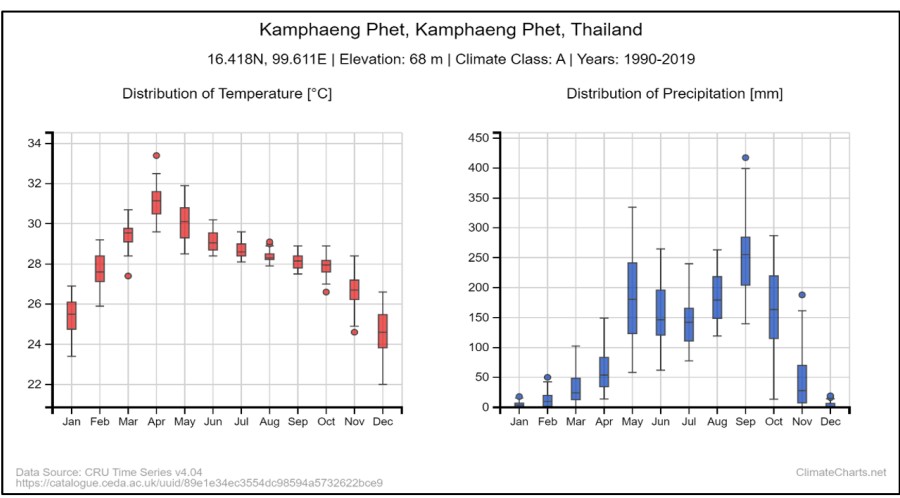

**Figure 1.** Annual temperatures and annual rainfall in Kamphaeng Phet Province [16].

Sugarcane has been popular with farmers in Kamphaeng Phet because sugarcane can resist intense storms better than other plants. Total sugarcane cultivation in the province covers 0.29 million acres. A defect detection model previously developed by the current authors is able to assess sugarcane defects caused by adverse conditions, such as sugarcane that collapses in a storm or sugarcane that does not grow due to drought (Tanut et al. [17]). That defect detection model based on the analysis of high-resolution color images obtained by the drone has an accuracy of 92.95% compared to expert evaluation.

The current study introduces a new sugarcane yield forecasting model called the Wondercane model, which is based on both sugarcane defect assessment and environmental factors. The defect assessment, which comes from analysis of field images obtained by drone, is used to account for unique characteristics of each sugarcane field that are caused by adverse conditions and which affects the final yield. The environmental factors, obtained from sugar mill field surveys and from government agencies, consist of rainfall, sugarcane variety, ratoon cut count (RCC), planting distance, and soil group. Another vital kind of data obtained from sugar mill surveys is the actual yield of past harvests. An important principle in the Wondercane model is that predicted yield is determined not through data in regression form (continuous variables) but rather through data in classification form (categorical variables). This is because the classification model takes a shorter modeling time than the regression model (since it does not require training data for several years), it can be predicted at the farming scale, and it is easily integrated with image analysis data. The feature extraction process of this model uses data mining and the concept of reverse design, together with the Similarity Relationship Method. This provides three important components necessary for the development of the model: Input Factors (IFs), Target Outputs (TOs), and Target Output relationships (TORs). The characteristics of the Wondercane model enable it to effectively reduce errors in yield assessment and optimize the allocation of sugar mill resources.

## 2. Materials and Methods

The Wondercane model is developed using three kinds of software: (1) MATLAB 2015B software to build and test the model, (2) Google Earth Editor for creation of virtual farm areas, and (3) ArcGIS software for mapping the farm into geographical areas. The hardware used is a the DJI Phantom 4 RTK (DJI-P4RTK) for capturing aerial photos. The DJI Phantom4 [18] is capable of flying for up to 30 min, with a wind resistance of 10 m/s, and it is equipped with an ultra-high-definition camera with a resolution of 20,962 cm$^2$ calculated from 72 dpi. Figure 2 shows a conceptual framework of the Wondercane pre-harvest sugarcane yield forecasting model with its four main processes: data collection, feature extraction, model creation and testing, and application development. Each of these processes is explained below.

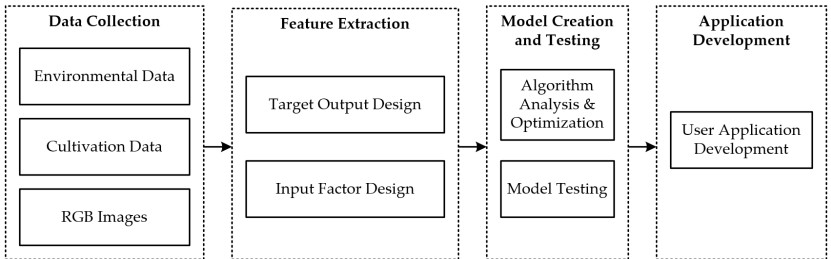

**Figure 2.** Conceptual framework of the Wondercane model.

## 2.1. Data Collection

The dataset for Wondercane comes from three sources: the local sugar mill that will harvest the cane, the Meteorological Department of Thailand (Ministry of Digital Economy and Society), and the Land Development Department (Thai Ministry of Agriculture and Cooperatives). Data on sugarcane variety, RCC, planting distance, and yield comes from the sugar mill. In this study the sugar mill was Nakornphet Sugar Company in Kamphaeng Phet City. Data on rainfall [19] and soil groups [20] were requested from the Meteorological Department and Land Development Department, respectively. In order to incorporate rainfall information into this study, the centroid (a Shapefile file format) of the sugarcane fields was mapped to the districts of Kamphaeng Phet Province using intersection tools of ArcGIS software [21]. In the same way, the centroid of the sugarcane fields also was mapped to the soil group data. The dataset which was collected during the harvesting phase in 2019 was used to train the model. Then the developed model was tested for its forecasting accuracy with the 2020 actual harvest dataset. A diagram of the data collection process is shown in Figure 3.

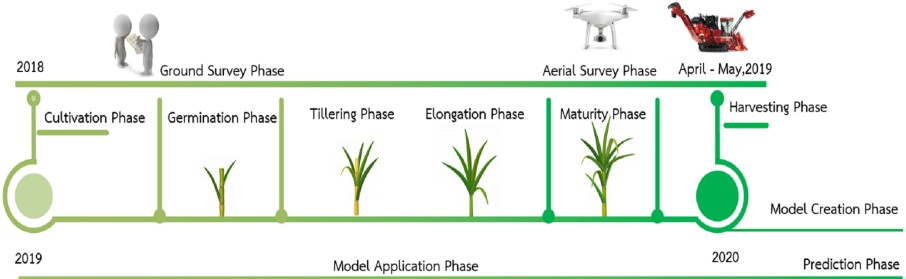

**Figure 3.** Diagram of the data collection process.

Images of the sugarcane field were collected from October to January in both 2019 and 2020, which are the months of the sugarcane maturity phase, when the drone can capture images showing defects in the field. The sampling images were collected and then analyzed for four different field factors: sugarcane variety, RCC, yield level, and soil group. The number of possible values in each of these field factors is as follows: 3 possible sugarcane varieties, 3 possible RCCs, 3 possible yield levels, and 22 possible soil groups. The Cartesian product [22] of the four field factors with all their possible values generated a total of 594 possible combinations of field factor values (i.e., $3 \times 3 \times 3 \times 22$). In this study, each possible combination of field factor values will be called a field profile. Ten examples of field profiles are shown in Table 1.

**Table 1.** Ten examples of field profiles.

| Example No. | Sugarcane Variety | Ratoon Cut Count | Yield Level | Soil Group |
|---|---|---|---|---|
| 1 | Lk-92-11 | 12 | Low | 40 |
| 2 | Khon Kaen 3 | 18 | Low | 40 |
| 3 | U Thong 11 | 12 | Low | 16 |
| 4 | Lk-92-11 | 12 | Medium | 15 |
| 5 | Khon Kaen 3 | 12 | Medium | 21 |
| 6 | Lk-92-11 | 18 | Medium | 6 |
| 7 | Lk-92-11 | 12 | High | 49 |
| 8 | Khon Kaen 3 | 15 | High | 36 |
| 9 | U Thong 11 | 12 | High | 38 |
| 10 | Lk-92-11 | 12 | High | 40 |

Figure 4 shows a silhouette of Kamphaeng Phet Province and its subdistricts, with red dots indicating the location of sugarcane fields surveyed by local sugar mills in 2018–2019 and 2019–2020.

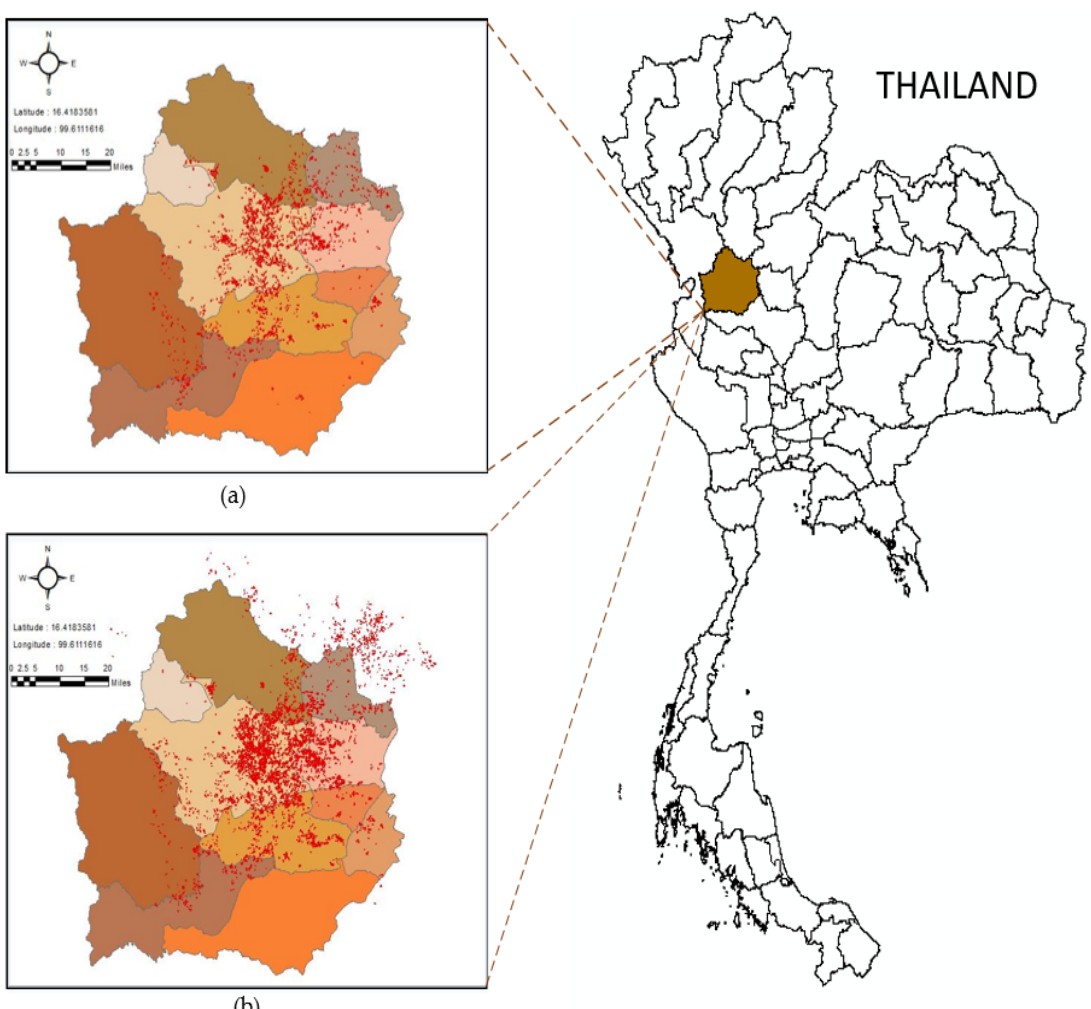

**Figure 4.** Silhouette of Kamphaeng Phet Province and its subdistricts, with red dots indicating the location of sugar cane fields surveyed by local sugar mills: (**a**) 2018–2019 and (**b**) 2019–2020.

To correctly survey the field, images need to be captured with three specifications: the drone must be 200–300 m above the ground, the images must be taken during daylight, and the resolution needs to be 3078 × 5472 pixels. After the aerial survey, it was found that

the 2018–2019 data produced 90 images, and the 2019–2020 data produced 72 images. To find the defect rate, all of these images were analyzed by the Sugarcane Defect Detection Program (SDDP) [17]. The sugarcane defect detection program's defect analysis results are shown in Figures 5 and 6.

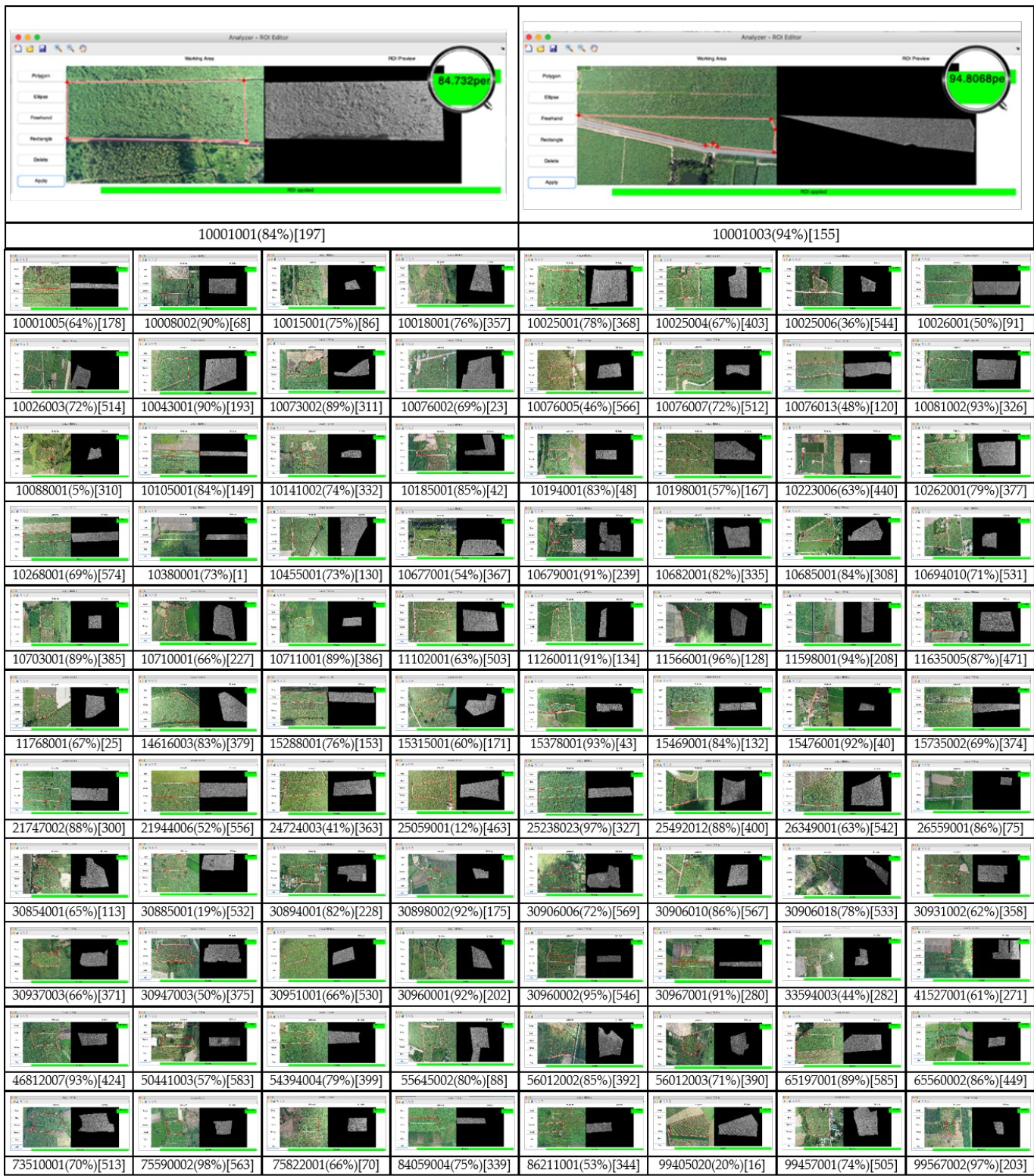

**Figure 5.** The sugarcane defect detection program's defect analysis for images in 2018–2019.

The text under each individual image in Figures 5 and 6 has three parts (from left to right): (1) the eight-digit code is the farm ID, (2) the number in parenthesis is the defect rate percentage, and (3) the number in brackets is the field profile of the image.

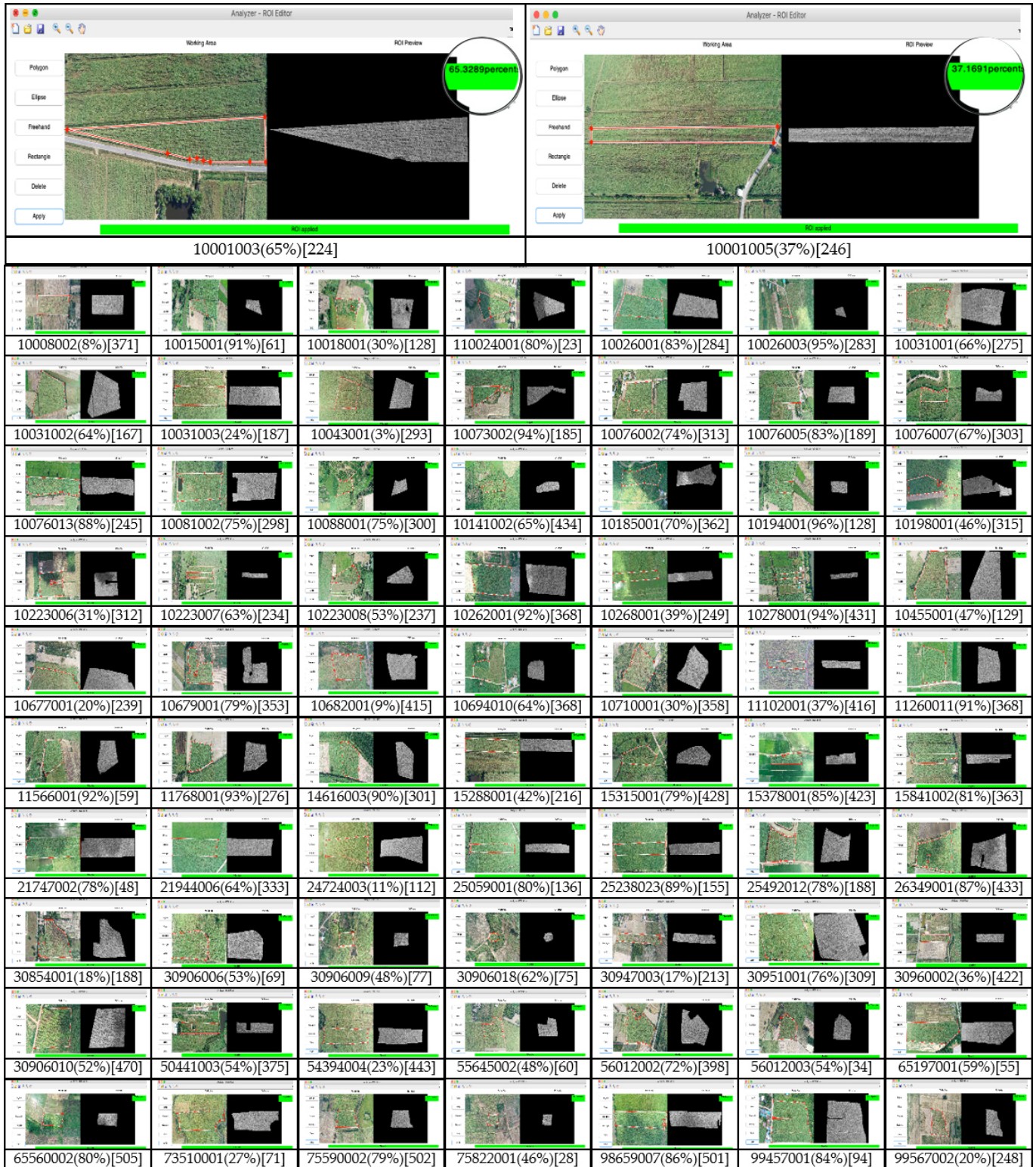

**Figure 6.** The sugarcane defect detection program's defect analysis for images in 2019–2020.

The current study's dataset includes environmental data and images collected during those surveys. Specific quantified information regarding the surveys and the resulting data used is shown in Table 2.

**Table 2.** Information on the field surveys and the resulting data used.

| C.P | No.O | | No.H (Fields) | C.R (Records) | Dataset | |
|---|---|---|---|---|---|---|
| | Ground (Fields) | Aerial (Fields) | | | T.V (Records) | Testing (Records) |
| 2018–2019 | 3442 | 90 | 1711 | 745 | 616 | 129 |
| 2019–2020 | 8117 | 72 | 2869 | 821 | - | 821 |

C.P: Cultivation period; No.O: No. of observations; No.H: No. of the harvested fields; C.D: Composite records; T.V: Training and Validation.

After the harvesting season, records from the environmental data of the surveys and records from the defect rate analysis of the aerial images were matched, and each matched pair of records was then merged to form only one combined record for each farm, called a composite record. It is these composite records that were used to develop the Wondercane model. A total of 745 composite records came from 2018–2019, of which 616 were used for model training and result validation (Raw Data Subset A) and the remaining 129 were used for testing (Raw Data Subset B). A total of 821 composite records came from 2019–2020, all of which were used for testing the model (Raw Data Subset C). All composite records were used for initial feature extraction.

### 2.2. Feature Extraction

In the Wondercane model, feature extraction of the datasets is accomplished by employing the concept of reverse design. The purpose here is to extract and optimize the Target Outputs (TOs), Target Output relationships (TORs), and input factors (IFs). A conceptual framework of this process is shown in Figure 7.

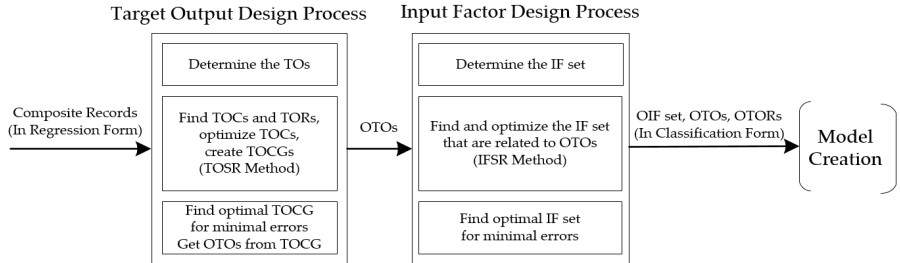

**Figure 7.** Conceptual framework of the feature extraction process (TO = Target Output, TOR = TO Relationship, TOC = TO Cluster, TOCG = TOC Group, IF = Input Factor).

The purpose of the feature extraction process is to determine two things: the Target Outputs and the Input Factors. The Target Output design process uses the six composite environment factors (rainfall, sugarcane variety, ratoon cut count (RCC), planting distance, soil group, and defect rate) to determine the TORs, based on similarity of the environmental factor profiles of all the TOs, and then finds the optimal set of TOs with the least errors. The results from the Target Output design process are used in the Input Factor design process, which will find the optimal set of IF set that are most related to the optimal TOs from the previous process. All three critical components (TOs, TORs, and IFs) resulting from the feature extraction process are then used to create the model. More specific information on each stage of the feature extraction process follows.

#### 2.2.1. The Target Output Design Process

In the first stage of the TO Design Process, the TOs are determined one by one using numbers with $2^n$, for example $2^1$, $2^2$, $2^3$, ..., $2^n$ where $n$ = 1, 2, 3, ..., $n$. It is important

to understand that as each TO is determined, it will travel through the entire TO Design Process from beginning to end before the next TO is then determined and follows the same path. As will be explained later, at one point it will become clear that the last useful TO has been reached. At that point, further TO determination will cease.

The second stage of the Target Output Design Process consists of many steps. All of these steps together constitute a new method called the Target Output Similarity Relationship (TOSR) Method, which is shown in Figure 8. During the TOSR method, all six composite record factors have to be considered: rainfall, soil group, planting distance, RCC, sugarcane variety, and defect rate.

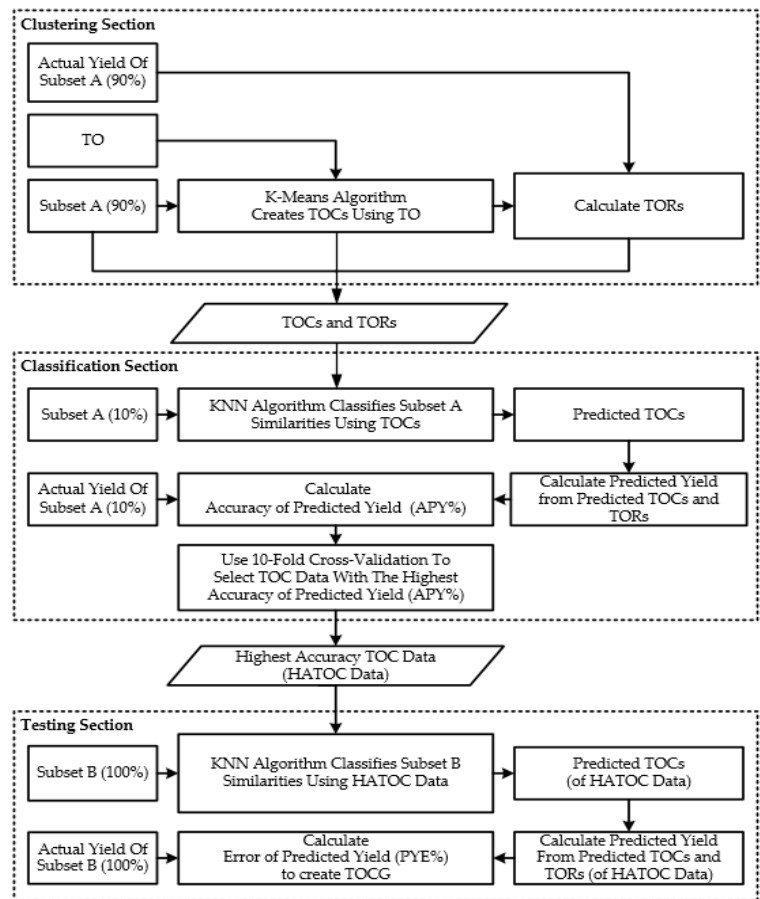

**Figure 8.** Conceptual framework of the Target Output Similarity Relationship Method (TO = Target Output, TOC = TO Cluster, TOR = TO Relationship).

In the Clustering Section of the TOSR Method, the K-Means Algorithm [23] is used to group the composite records (specifically 90% of Raw Data Subset A) around the one determined TO that is traveling through. This creates TO Clusters (TOCs) for that TO. The TORs are then calculated from the average actual yield of each TOC.

In the Classification Section of the TOSR Method, the K-Nearest Neighbor Algorithm [24] (with Neighbor set at 3) is run to classify the similarities of the composite records in 10% of Raw Data Subset A using the TOCs. The result is Predicted TOCs. Those Predicted TOCs are then used with the TORs from the Clustering Section to calculate Predicted Yield. Next, the Predicted Yields are compared with the actual yields to calculate the Accuracy of Predicted Yield (APY%). At the end of the Classification Section of the TOSR Method, 10-fold cross-validation [25] is used to select the TOC data with the highest Accuracy of Predicted Yield (APY%), hereafter referred to as the Highest Accuracy TOC data (HATOC data). These can be thought of as optimized TOCs.

The Testing Section is the last part of the TOSR Method. Here, the KNN Algorithm classifies the similarities of the composite records (Raw Data Subset B) using the HATOC data to obtain new Predicted TOCs. Predicted Yield is again calculated, and finally the Predicted Yield is compared to Raw Data Subset B Actual Yield to determine the Predicted Yield Error rate (PYE%). As a result, a TOC Group (TOCG) is created. Note that one original TO became one TOCG. The TOCG contains a list of TOCs, and the TOCG is identified by the number of TOCs in its list. For example, TOCG-64 has 64 TOCs inside. Thus the TOCG has qualities of list, but it also has qualities of a counter, indicating how many TOCs it represents. Creation of one TOCG for each TO traveling through is the ultimate goal of the TOSR Method. Thus ends the TOSR Method and also the second stage of the TO Design Process.

In the third stage of the TO Design Process, the TOCG identifying number (i.e., 2 for TOCG-2, 4 for TOCG-4, 8 forTOCG-8, etc.) of each TOCG that arrives from the previous stage (i.e., from the TOSR Method), along with its corresponding PYE rate (each TOCG has one PYE%), is plotted in a cumulative trend graph where each arriving TOCG will be added. Two big tasks remain here in the third stage. First, we must figure out when to stop adding new TOs (and new TOCGs). Second, we must find the one optimal TOCG that will result in a minimum error rate. Since one aspect of a TOCG is to function as a counter of the TOCs inside it, determining the optimal TOCG will also reveal how many TOCs will be used for optimum effect. However, as mentioned we must first look for the right moment to stop adding new TOs (and new TOCGs). As each TOCG arrives and is plotted in the cumulative trend graph, there will come a time when the latest TOCG to arrive will have a PYE% that is dramatically higher than the others. That newly arrived TOCG with a dramatically higher PYE% is the sign that this TOCG is the last TOCG to include in the graph, and thereafter no more TOs should be inserted into the beginning of the TO Design Process. The graph is now full.

Once the graph is ready, the Bisection Method [26] is then applied to find the optimal TOCG. This means first identifying the lowest predicted yield error rate (PYE%) in the current graph and then narrowing the range, with a focus on that current lowest PYE%. An abstract figure of the Bisection Method is shown in Figure 9. For example, when the lowest PYE in the graph is found to be at f(x3), then the range limits will be narrowed to be closer to f(x3) point on both sides, left and right. In this example, the new lower range limit (on the left) is calculated as the midpoint between x2 and x3, and the upper range limit (on the right) is calculated as the midpoint between x3 and x4. After the range is thus narrowed, the PYE% is re-calculated using the new TOCG identifying numbers from the new narrower range. By repeating this process multiple times, the progressively narrower range eventually converges on the optimal TOCG. This process ends when there is insufficient range for further division, at which time the optimal TOCG is taken as the TOCG identifying number with the lowest PYE% in the final iteration of the graph. Once the optimal TOCG is in hand, the TOCs inside that TOCG will be called the Optimized TOs (OTOs), and these OTOs will be used going forward.

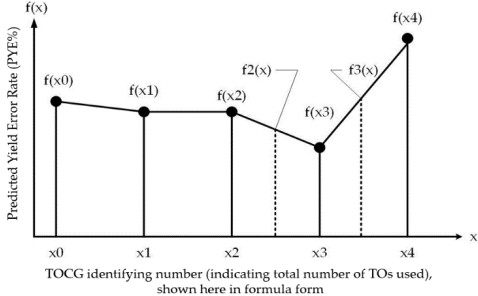

**Figure 9.** An abstract example of the Bisection Method in the trend graph for determining the optimum TOCG having minimum errors.

### 2.2.2. The Input Factor Design Process

In the first stage of the IF Design Process, the Input Factor sets are determined by finding the power set of all six composite record factors (i.e., rainfall, soil group, planting distance, RCC, sugarcane variety, and defect rate) using the subset method [27]. The result is 32 IF sets with no empty set, and there is a defect rate in each set. For example, IF set#1 = {defect rate}, IF set#2 = {rainfall, defect rate} IF set#3 = {soil groups, defect rate}, ..., IF set#32 = {rainfall, soil groups, planting distance, RCCs, sugarcane varieties, defect rate}. Every IF set contains composite records, the same total number of composite records being processed. Each of the 32 IF sets will go through the rest of the IF Design Process one by one. All 32 sets will go through: no more, no less.

The second stage of the Input Factor Design Process has some pathways that are analogous to pathways in the second stage of the previous Target Output Design Process and its TOSR Method. However, many important components are different here, so a separate Input Factor Similarity Relationship (IFSR) Method is used here, shown in Figure 10.

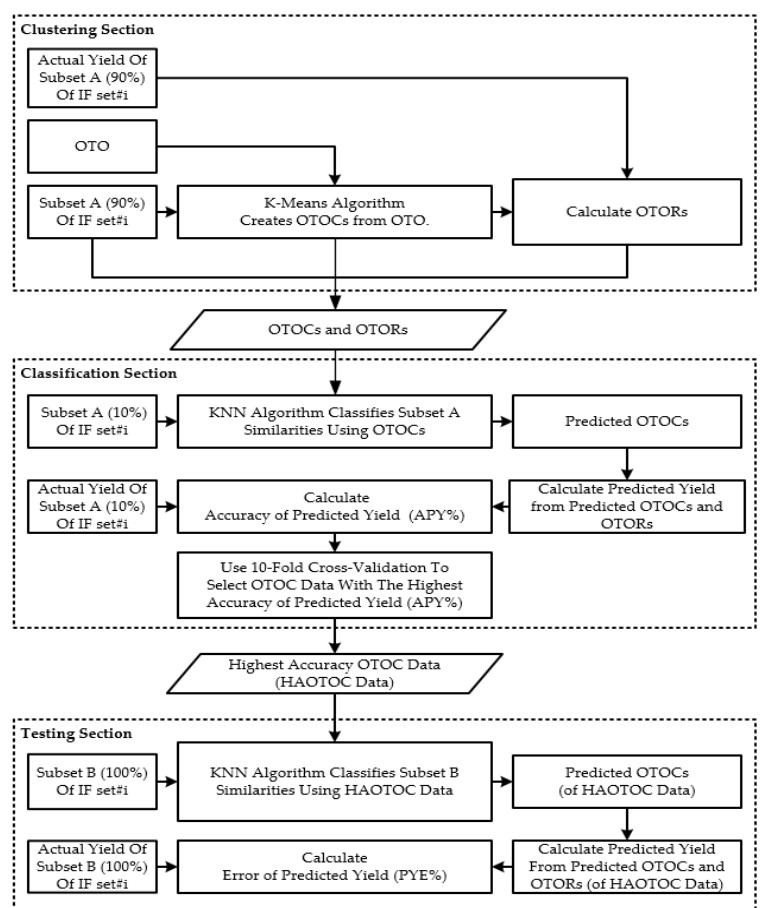

**Figure 10.** Conceptual framework of the Input Factor Similarity Relationship Method (OTO = Optimal Target Output, OTOC = Optimal TO Cluster, OTOR = Optimal TO Relationship).

In the Clustering Section of the IFSR Method, the K-Means Algorithm is used to group the composite records (sourced from 90% of Raw Data Subset A) of each IF set around the one OTO (from the TO Design Process) to create OTO Clusters (OTOCs). Although the IF set changes each time here, the one OTO does not change. The OTO Relationships (OTORs) are then calculated from the average actual yield of each OTOC.

In the Classification Section of the IFSR Method, the K-Nearest Neighbor Algorithm (with Neighbor set at 3) is run to classify the similarities of the composite records of each IF set in 10% of Raw Data Subset A using the OTOCs. The result is Predicted OTOCs. Those

Predicted OTOCs are then used with the OTORs from the Clustering Section to calculate Predicted Yield. Next, the Predicted Yields are compared with the actual yields to calculate the Accuracy of Predicted Yield (APY%). At the end of the Classification Section of the IFSR Method, 10-fold cross-validation is used to select the OTOC data with the highest Accuracy of Predicted Yield (APY%), hereafter referred to as the Highest Accuracy OTOC data (HAOTOC data).

The Testing Section is the last part of the IFSR Method. Here, the KNN Algorithm classifies the similarities of the composite records of each IF set (Raw Data Subset B) using the HAOTOC data to obtain new Predicted OTOCs. Predicted Yield is again calculated, and finally the Predicted Yield is compared to Raw Data Subset B Actual Yield to determine the Predicted Yield Error rate (PYE%). This is the end of the IFSR Method and also the end of the second stage of the IF Design Process.

In the third stage of the IF Design Process, the purpose is find the one optimal IF set that will result in the minimum predicted yield error rate. To find that optimal IF set, all the IF sets (i.e., IF set#1, IF set#2, IF set#3, . . . , IF set#32) and corresponding PYE rates (each IF set has one PYE%) from the end of the previous stage are first plotted in a bar graph. Then the IF set that has the lowest PYE% in the graph is selected. This will be called the Optimized IF (OIF) set. The OIF set, OTOs, and OTORs in essence represent the original Raw Data Sets (A, B, and C) that have now been transformed through classification, so those will now be referred to as Classified Data Sets A, B, and C. All necessary information is now ready for creation and testing of the Wondercane model.

### 2.2.3. Model Creation and Model Testing

The Model Foundation Data specifically from the 2019 harvest (745 records) was used to trial six candidate algorithms in order to compare the effectiveness of these algorithms at recognizing OTOCs and accurately predicting yield. This will identify the best algorithm to ultimately select for model creation. The six algorithms in this experiment were: K-Nearest Neighbor (KNN) (K = 3) [28], Random Forest (FOR) (batch size = 100) [29], Random Tree (RTR) (batch size = 100) [30], Reduced Error Pruning Tree, a.k.a. REP Tree (REP) (batch size = 100) [31], Decision Tree (DEC) (batch size = 100) [30], and Multilayer Perceptron (MLP) (ANN = 5:18:32, L = 0.3, M = 0.2, Epoch = 500) [32]. Each of these algorithms was used separately in turn to train and test the model, as follows. First, Classified Data Subset A was used with the 10-fold validation method to both train and test the model. Nine folds (subsamples) were used for training and one fold was used for testing. After that, Classified Data Subset B was then used to further test the model using the blind test method [33]. WEKA software [34] is the environment where these training and testing procedures were carried out in search of the algorithm with the best OTO recognition and the closest actual yield prediction. Once the optimal algorithm was determined, the completed model was used on Classified Data Subset C as an additional, final test to confirm that the model works correctly on any data from any year. The results obtained from this final test will verify how effective the model is at forecasting yield.

### 2.2.4. Application Development

The Wondercane model will be developed into a user application that can run on any company's internal website. Wondercane works in conjunction with the Sugarcane Defect Detection Program (SDDP) previously developed by the current authors [17], because defect percentages are required to characterize the environment and cultivation issues of the sugarcane fields being analyzed.

## 3. Results

### 3.1. Target Output Design Process Experiment Results

After the TOCGs and their corresponding PYE rates were determined in Stage 2 of the Target Output Design Process, the TOCGs were loaded one by one into a trend graph, looking out for a dramatic increase in PYE%, which would be the signal that all relevant

TOCGS had been loaded into the graph and new TO determination should stop. That dramatic increase in PYE% happened where n = 9, corresponding to the addition of TOCG-512. Figure 11 shows that initial instance of the fully populated trend graph. After that, the bisection method was applied to zoom in on the exact TOCG with the lowest PYE%.

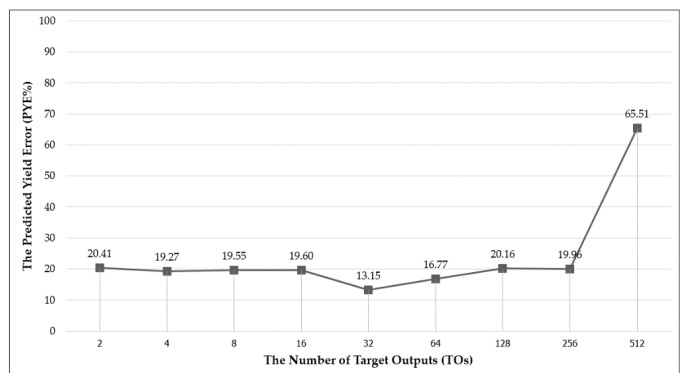

**Figure 11.** Initial instance of the fully populated trend graph of TOCG numbers and corresponding PYE rates, before the start of the bisection method.

Table 3 summarizes the results of applying the bisection method to the fully populated trend graph. There were a total of six rounds. Each round represented a consecutive narrowing of the range in the trend graph.

**Table 3.** Results of the bisection method used to complete the last part of the TO design process.

| Round | Starting Range | Ending Left TO | Ending Right TO |
|---|---|---|---|
| 1 | 2,4,8,16,32,64,128,256,512 | 16 | 64 |
| 2 | 16,32,64 | 24 | 48 |
| 3 | 24,32,48 | 28 | 40 |
| 4 | 28,32,40 | 30 | 36 |
| 5 | 30,32,36 | 31 | 34 |
| 6 | 31,32,34 | 32 | 33 |

In the first round of bisection, the initial instance of the trend graph (Figure 11) suggested that 32 TOs (TOCG-32) with a PYE% of 13.15 had the minimal error rate when compared with the other options. However, we cannot know for sure what the optimal number of TOs is without first drilling down to the left and right of the current minimum PYE% to calculate the PYE%s of the other nearby TO numbers. A close neighbor might actually have a lower PYE% than 32 TOs. The starting range of Round 1 has too many possible TOCGs to narrow the range using a formula (because the formula requires that there be only 3 TOCGs), so just for Round 1 the new narrower range of 16-64 TOs (new ending left TO = 16, new ending right TO = 64) was selected visually, based on the current lowest PYE% (32 TOs) and the TOCGs directly to the right and left. The PYE%s were then recalculated for that new narrower range before proceeding to Round 2.

Beginning in Round 2, the range was narrowed by formula. Specifically, the new Ending Left TO was determined by this calculation: (16 + 32)/2 = 24 (i.e., 24 TOs). The new Ending Right TO was determined by this calculation: (32 + 64)/2 = 46 (i.e., 46 TOs). Each subsequent round proceeded similarly, except that in Round 6, only the Ending Right TO was calculated by formula, because on the left side (31 + 32)/2 = 31.5, and only whole number answers are applicable. Round 6 was the final round, because at the end of Round 6 there was nothing left to bisect. Figure 12 shows the previously known PYE%s plus all the new PYE%s gleaned from 6 rounds of range narrowing.

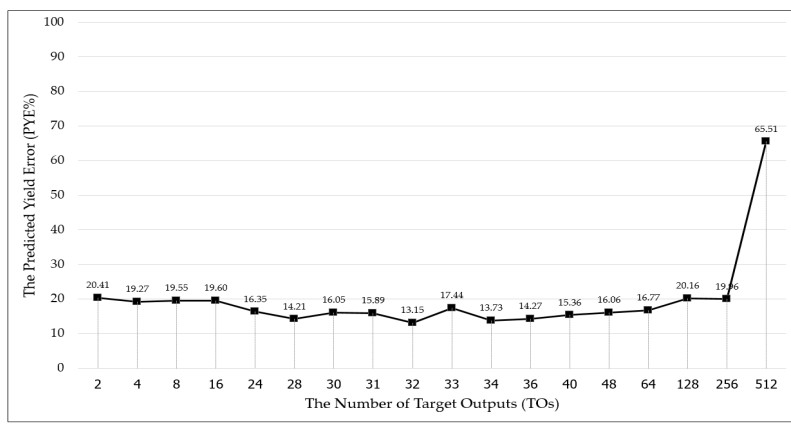

**Figure 12.** All PYE%s resulting from the TO design process.

The results of the TO design process experiments revealed 32 TOs to be the best solution, yielding the lowest predicted yield error rate (PYE%) of 13.15%. Therefore, these 32 optimized TOs (OTOs) were sent to the Input Factor Design Process for the subsequent experiments.

### 3.2. Input Factor Design Process Experiment Results

At the final culmination of the Input Factor Design Process, all the IF sets (i.e., IF set#1, IF set#2, IF set#3, ..., IF set#32) and their corresponding PYE rates from the previous stages of the IFDP were plotted in a bar graph in order to find the optimal IF set with the lowest PYE rate. The completed graph is shown in Figure 13.

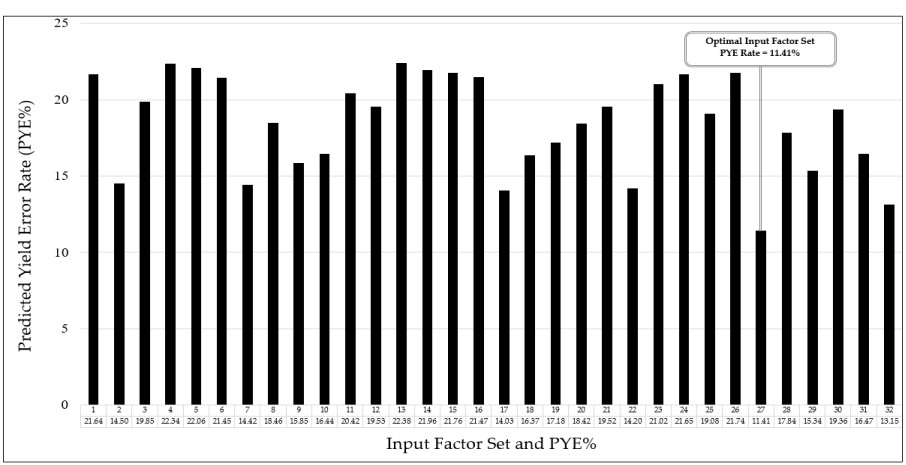

**Figure 13.** IF sets and PYE rates from the IF Design Process. IF set#27 had the lowest PYE rate.

Of the 32 IF sets, the best result came from IF set#27, which had the lowest PYE rate at 11.41%. The IF set#27 contains five factors: rainfall, soil group, planting distance, RCC, and defect rate. (Sugarcane variety is not part of this particular IF set). These five factors were therefore used in the model creation process.

### 3.3. Model Creation Process Experiment Results

When the six candidate algorithms were trialed using WEKA Program during model creation in order to compare their effectiveness at recognizing OTOs and accurately predicting yield, the six confusion matrices shown in Figure 14 were produced. These confusion matrices represent the effectiveness of each algorithm at recognizing OTOCs.

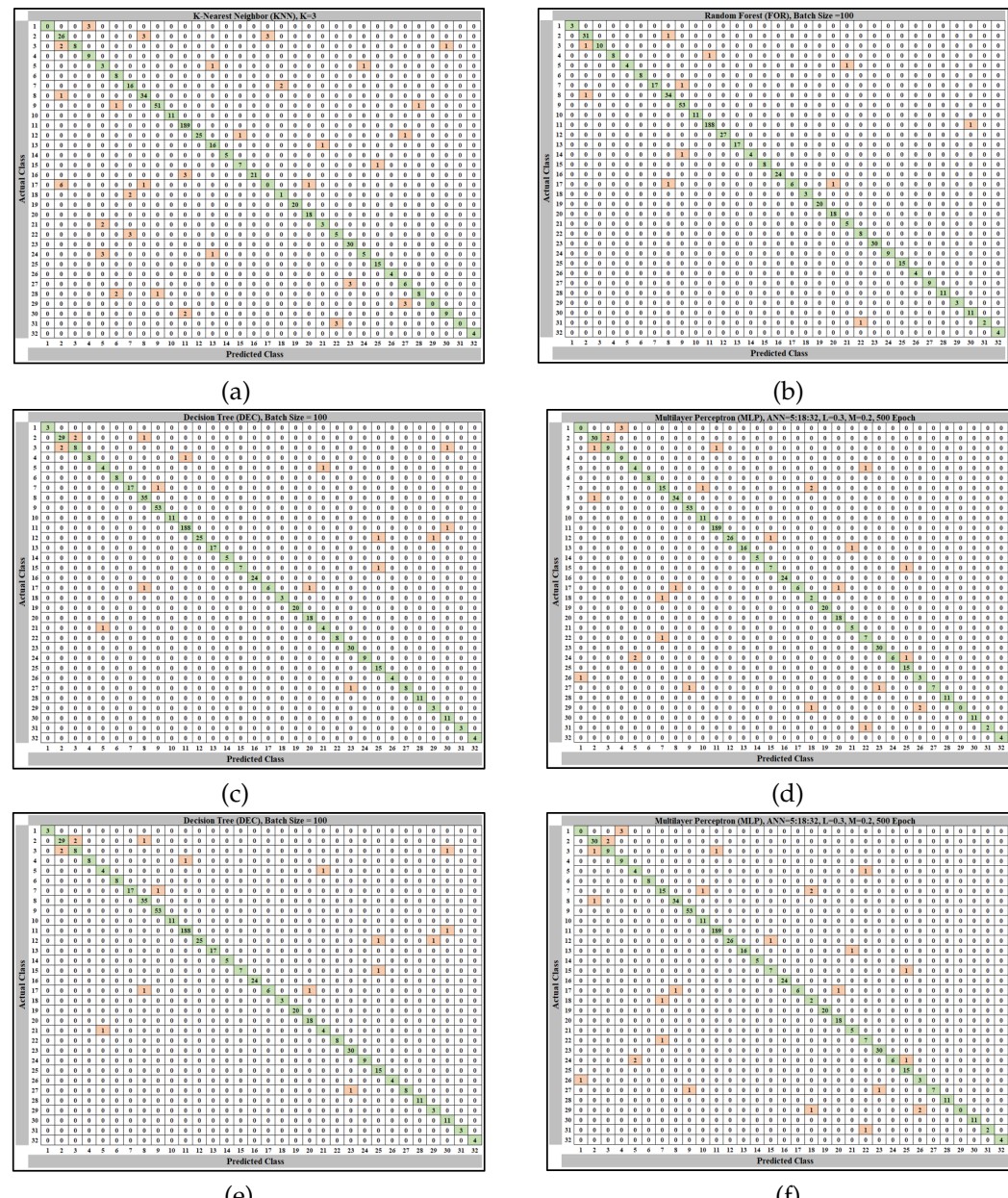

**Figure 14.** The confusion matrices resulting from the six trialed algorithms: (**a**) K-Nearest Neighbor, (**b**) Random Forest, (**c**) Random Tree, (**d**) Reduced Error Pruning (REP) Tree, (**e**) Decision Tree, and (**f**) Multilayer Perceptron.

After all of the algorithms in Figure 14 were tested using the Classified Data Subset A, two efficiency indicators (Overall Accuracy (OA) and Root Mean Squared Error (RMSE)) were calculated from the confusion matrix data of each of algorithm, and the results are shown in Table 4.

**Table 4.** The efficiency indicator results of each algorithm used during model creation.

| Indicators/Algorithms | KNN | FOR | RTR | REP | DEC | MLP |
|---|---|---|---|---|---|---|
| OA | 90.42 | 98.2143 | 97.4026 | 93.9935 | 97.2403 | 95.2922 |
| RMSE | 0.06 | 0.0326 | 0.0403 | 0.0533 | 0.0396 | 0.047 |

The efficiency indicator results revealed that the Random Forest (FOR) algorithm had the best performance among the six algorithms. This is because Random Forest had the

highest OA (98.2143) and the lowest RMSE (0.0326) of all six algorithms. The details of the Overall Accuracy indicator of the Random Forest algorithm were as follows: True Positive (TP) Rate = 0.983, False Positive (FP) Rate = 0.002, Recall = 0.982, F1-Measure = 0.982, and Receiver Operating Characteristics (ROC) Area = 0.998. The fact that the ROC Area was close to 1.00 shows that the Random Forest algorithm recognized the relationships between the OIFs and the OTOs with high precision. In order to validate these results via an alternative method, these five figures were recalculated using 1000 bootstraps. All five results from bootstrapping matched the original results. The Wondercane model was therefore constructed with the Random Forest algorithm at its heart. After that, the model was tested using Classified Data Subset B. WEKA software indicated that the model classified the TOs of Classified Data Subset B with an accuracy of 95.42%. Next, the predicted yields of Classified Data Subset B were determined by the model, and these were compared with the actual yields. The results of that experiment are shown in Figure 15.

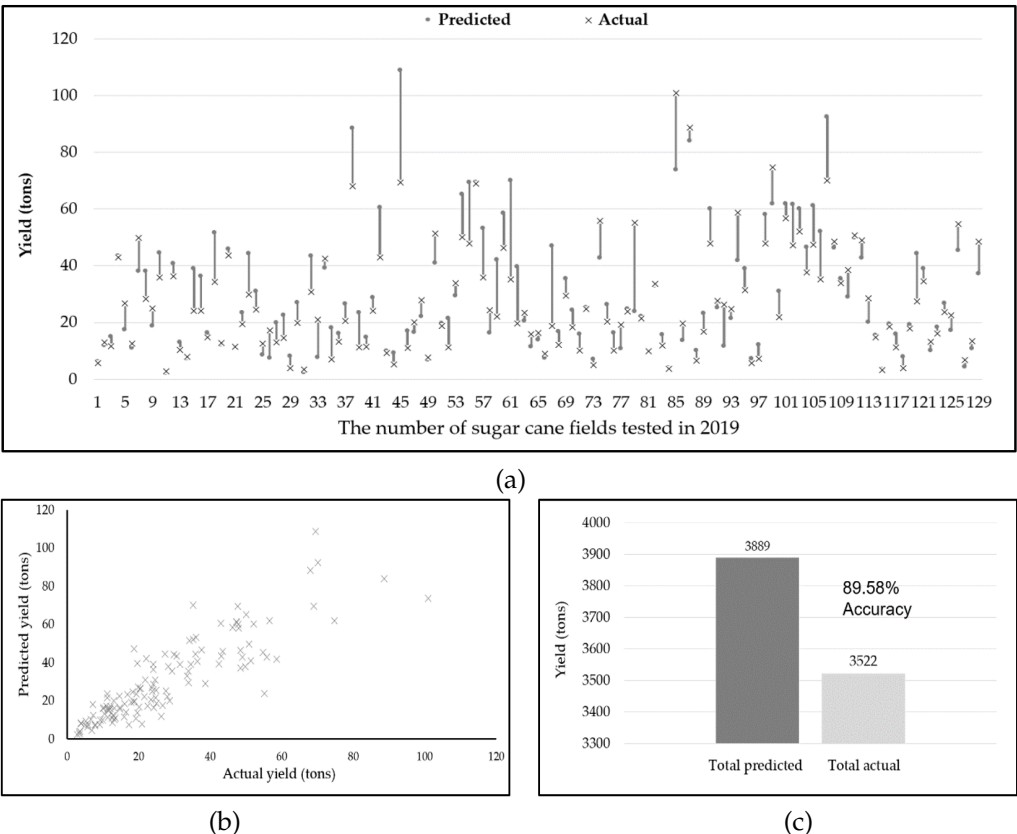

**Figure 15.** Test results when the finished model was run on Classified Data Subset B: (**a**) The difference between each predicted yield and actual yield is displayed as a line. (**b**) The correlation between predicted yield and actual yield. (**c**) Total predicted yield compared with total actual yield.

When the Correlation Coefficient was calculated between the predicted and actual yields across all 129 sugar fields, the value was R = 0.88 ($p < 0.01$), indicating a strong positive correlation that is statistically significant. The model predicted the total yield with an accuracy of 89.58% across the total area of 871 acres.

Finally, the Wondercane model was re-tested with Classified Data Subset C (821 Records) in order to confirm its accuracy. The experiment results are shown in Figure 16.

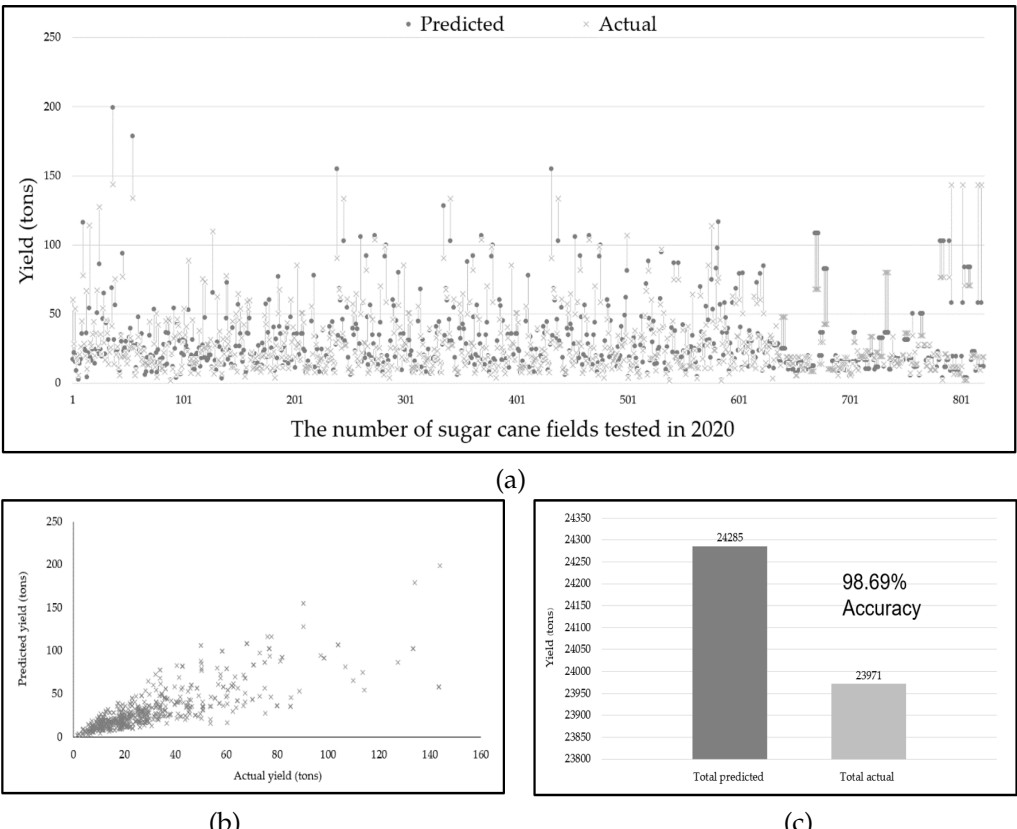

**Figure 16.** Test results when the finished model was run on Classified Data Subset C: (**a**) The difference between each predicted yield and actual yield is displayed as a line. (**b**) The correlation between predicted yield and actual yield. (**c**) Total predicted yield compared with total actual yield.

When the Correlation Coefficient was calculated between the predicted and actual yields this time across all 821 sugar fields, the value was R = 0.80 ($p < 0.01$), very similar to the previous R = 0.88 for Classified Data Subset B. This correlation coefficient value again indicates a strong positive correlation that is statistically significant. The model predicted the total yield with an accuracy of 98.69% across the total area of 5483 acres.

### 3.4. Application Development

The Wondercane model was developed into a user application that can run on any company's internal website. Figure 17a shows the main screen of the Wondercane Sugarcane Yield Forecasting Program. Figure 17b shows the companion Sugarcane Defect Detection Program (SDDP), which is used to upload the user's sugarcane field image and select the area of the field to analyze for defects.

To begin, the user opens the SDDP and on its main screen uploads the drone image of the sugarcane field that they wish to analyze. Then using a selection tool, the user indicates to the program which portion of the image to analyze for defects. Objects like trees can be excluded from the selection. Once the selection is ready, clicking the Process button initiates defect analysis. The defect results are then displayed as an area percentage in the upper right corner of the program window. After that, the Upload button saves the results and sends them to the Wondercane program, which opens automatically in a new tab. In the Wondercane program, the user fills in the requested environmental data and the total area. The final yield forecast is then displayed in two formats: as Yield Rate (tons per acre), and as Total Yield (tons).

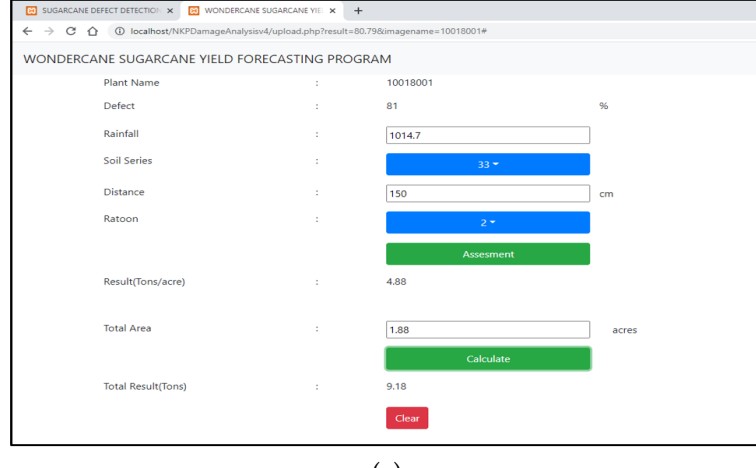

(a)

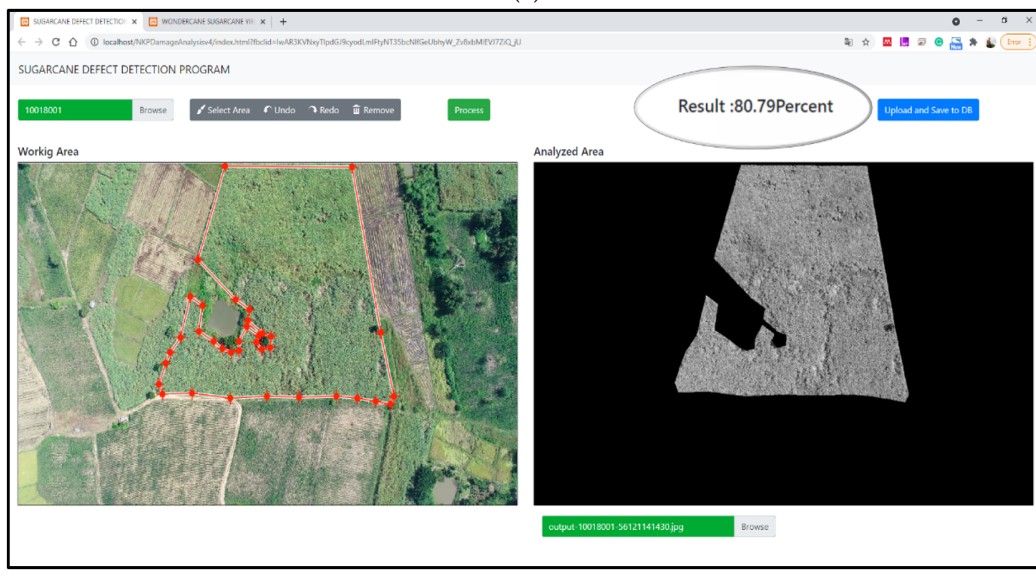

(b)

**Figure 17.** Sample screenshots of (**a**) the Wondercane sugarcane yield forecasting program working with (**b**) the sugarcane defect detection program.

## 4. Discussion

The results of the Wondercane forecast yields were well aligned with the actual yield, providing an effective level of accuracy. The model's strengths lie in how cultivation issues are incorporated into the defect rate and how the reverse design method is used to find the optimal target outputs, input factors, and classification method. The Wondercane results also stand up well against results of the previous studies mentioned in the introduction. An overview of the Wondercane model in comparison to ten previous studies, looking at key features as well as final correlation and accuracy, is shown in Table 5.

**Table 5.** Overview of the Wondercane model in comparison to ten previous studies, looking at key features as well as final correlation and accuracy.

| Study | Crop | Scale | FC | IFs | Images | Classifier | R$^2$ | OA(%) |
|---|---|---|---|---|---|---|---|---|
| Wondercane: | | | | | | | | |
| | Sugarcane | Farming | 821 | 5 | RGB | RF | 0.80 | 98.69 |
| Previous Studies: | | | | | | | | |
| [6] | Sugarcane | Region | - | 8 | - | ANN | - | 99.80 |
| [7] | Sugarcane | Region | - | 18 | - | RF | 0.43 | 66.00 |
| [8] | Sugarcane | Region | - | 9 | - | AES | - | 94.01 |
| [9] | Corn | Region | - | 7 | Satellite | ANN | 0.75 | 93.79 |
| [10] | Sugarcane | Region | - | 7 | Satellite | LAI | 0.88 | 91.70 |
| [11] | Sugarcane | Region | - | 1 | Satellite | GNDVI | 0.69 | 69.00 |
| [12] | Vineyard | Farming | 3 | 7 | Drone: IF+RGB | CWSI+NDVI | 0.69 | 80.00 |
| [13] | Sugar beet | Farming | 1 | 7 | Drone: IF+RGB | CHM+VI | - | 80.00 |
| [14] | Sugarcane | Farming | 2 | 2 | Drone: RGB | OBIA+GLCM | - | 92.00 |
| [15] | Sugarcane | Farming | 15 | 2 | Drone: RGB | GRVII+LAI | 0.79 | 90.00 |

FC = Field Count, IFs = Input Factors, R$^2$ = Correlation between predicted yield and actual yield, OA = Overall Accuracy, RF = Random Forest, ANN = Artificial Neural Network, AES = Adaptive Evolution Strategies, LAI = Leaf Area Index, GNDVI = Green Normalized Difference Vegetation Index, CWSI = Crop Water Stress Index, CHM = Canopy Height Model, VI = Vegetation Index, OBIA = Object-Based Image Analysis, GLCM = Grey-Level Co-occurrence Matrix, GRNI = Green-Red Vegetation Index.

Each of the studies compared in Table 4 was based on its own particular environment and objectives, thus the key features vary. These studies can be divided into two groups: studies based solely on environmental data ([6–8]), and studies based on environmental data as well as images ([9] through [15]). Using images is a way to assess crop defects, so the first group of previous studies lacks the benefit of this perspective, while the second group is able to account for crop defects such as cane that collapsed during adverse weather, stunted growth due to excessive weeds, or a variety of other possible cultivation issues. The regional scale studies that involved images ([9–11]) used satellite images to assess plant diseases and monitor plant growth. In contrast, when the scale of the study was an individual farm ([12–15]), drone images were used instead for several reasons. The cost of operating a drone is much lower than the price of satellite images. A drone also has easier access to a field for closer range images. Finally, satellite images are vulnerable to cloud cover, while the drones fly below the clouds. Matese et al. [12] and Mink et al. [13] used drones equipped to provide not only standard RGB images but also infrared images. However, the added cost of the infrared capability did not result in increased forecast accuracy. Som-ard et al. [14] and Sanches et al. [15] used less expensive drones delivering only RGB images, and the forecast accuracy of their studies was higher than the studies that also used infrared images. The studies by Som-ard et al. [14] and Sanches et al. [15] both share similarities with the current study. They applied data from RGB color image analysis to estimate sugarcane yield. Som-ard's study resulted in a highly accurate forecast; however, the amount of data analyzed was small. On the other hand, Sanches' study used somewhat more data, but its forecast error was higher. In the current study, the Wondercane model used much more data than either the Som-ard or Sanches study, and the overall accuracy was higher than either of those studies. Wondercane stands out for its ability to incorporate data on adverse weather, defects such as damaged cane, and other cultivation issues. As a result, the model is able to forecast the total yield from 2020 data with an accuracy of 98.69%. Looking forward, there is also no reason why the Wondercane model cannot be further adapted and applied to other farm crops that have a physical structure similar to sugar cane, such as corn or cassava.

## 5. Conclusions

By using data mining and the developed reverse design method to integrate environmental data from sugar mills and government agencies with the analysis of aerial images

from drones, the new Wondercane model is able to correctly forecast total sugarcane harvest yield with an accuracy of 98.69%. The Wondercane model is therefore an accurate and robust tool that can substantially reduce the issue of sugarcane yield estimate errors and provide the sugar industry with improved pre-harvest assessment of sugarcane yield.

**Author Contributions:** Conceptualization, B.T. and P.R.; methodology, B.T.; software, B.T.; validation, B.T. and P.R.; formal analysis, B.T. and R.W.; investigation, P.R.; resources, B.T.; data curation, B.T.; writing—original draft preparation, B.T. and P.R.; writing—review and editing, B.T. and P.R.; visualization, B.T. and R.W.; supervision, P.R.; project administration, B.T. and P.R.; funding acquisition, B.T. and P.R. All authors have read and agreed to the published version of the manuscript.

**Funding:** This research received no external funding.

**Institutional Review Board Statement:** Not applicable.

**Informed Consent Statement:** Not applicable.

**Data Availability Statement:** The data can be found from the correspondence authors.

**Acknowledgments:** The authors would like to thank Nakornphet Sugar Co. Ltd. in Kamphaeng Phet City as well as Computer Vision and Human Interaction Technologies Laboratory at Naresuan University (NU Vision Lab) for their kind assistance with this project. Thank you also to Paul Freund of Naresuan University Writing Clinic (DIALD) for editing this manuscript.

**Conflicts of Interest:** The authors declare no conflict of interest.

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
