# Peer review of "High Accuracy Pre-Harvest Sugarcane Yield Forecasting Model Utilizing Drone Image Analysis, Data Mining, and Reverse Design Method"

_agriculture, doi:10.3390/agriculture11070682_

Round 1
Reviewer 1 Report
The manuscript presents a method for forecasting the sugarcane yield that should yield greater accuracy and could be used in a pre-harvest assessment.
The topic is very interesting as sugarcane is an important crop commodity in Thailand and accurate yield predictions will be very useful for pre-harvest resource allocation. We know that deep learning technologies using satellite imagery and various environmental datasets can be very useful for yield estimation and forecasting, as also presented by the authors in the discussion section.
Major comment: After reading the methodology and results section for several times, I still don't understand what methodology was applied and what problems were solved by introducing the workflow. The manuscript will need a lot more explanation on all the different components presented in the methodology before scientists that are not familiar with this work will understand why and how the different components were used and combined. It remains unclear how the drone imagery was processed and used, what the soil dataset are that go into the model, how rainfall information was included, what the uncertainties are in the different datasets, what the importance of the data in the overall model is etc. As such, I'm unable to judge the scientific soundness, significance and novelty of the manuscript.
Minor comments: - The Application Development section is of minor interest to a scientific audience. - The use of scientific literature is relatively weak. We have a short reference list of which 15 come from a single table. - The discussion section could use some more body with a deeper reflection on the applied methodology with strengths and weaknesses. - Figure 3 needs lat/lon information and it would be nice to have an image of Thailand with the locations of the province. - Comparing total estimated with total measured yield as an accuracy measurement is rather ambiguous.
Author Response
Dear Reviewer 1
We really appreciate your comments. We truly believe that your comments can improve our manuscript to be more interesting and more understanding for readers on the data collection and the discussion parts. My responses on each of your comments can be seen in the attachment.
Sincerely,
Assoc. Prof. Panomkhawn Riyamongkol

Reviewer 2 Report
This study integrated the environmental data and drone images to forecast the sugarcane yield, steps in the forecasting method were designed well; however my main concerns are:
(i) when this model (RF model) is applied to any other environment (untested or unknown environment), i am not fully convinced that model performance is gong to be same as the current results for the yield prediction,
(ii) genetic background of sugarcane variety is very diverse, I have not seen any predictor(s) related to characteristics (trait) of sugar cane variety as an input in the model, for instance temporal heights (canopy height measurement from drone images), flowering times etc. can be used as an input in the forecasting model that has been disregarded but needs to be considered for robust yield forecasting model.
My other comments are attached inside the PDF.

Author Response
Dear Reviewer 2
We really appreciate your suggestions. We believe that our manuscript cannot be better without your comments. The responses according to your comments have been attached in the attachment.
Sincerely,
Assoc. Prof. Panomkhawn Riyamongkol

Round 2
Reviewer 2 Report
Authors responded to my question and concerns, I have no any further concerns. Thanks.